# Studies on Anion Exchange Membrane and Interface Properties by Electrochemical Impedance Spectroscopy: The Role of pH

**DOI:** 10.3390/membranes11100771

**Published:** 2021-10-10

**Authors:** Wenjuan Zhang, Wei Cheng, Ramato Ashu Tufa, Caihong Liu, David Aili, Debabrata Chanda, Jing Chang, Shaopo Wang, Yufeng Zhang, Jun Ma

**Affiliations:** 1Tianjin Key Laboratory of Aquatic Science and Technology, School of Environmental and Municipal Engineering, Tianjin Chengjian University, Tianjin 300384, China; changjinghit@163.com (J.C.); wspfr@sina.com (S.W.); zyf9182@163.com (Y.Z.); 2Research Center for Eco-Environmental Sciences, Chinese Academy of Sciences, Beijing 100085, China; weicheng@rcees.ac.cn; 3Department of Energy Conversion and Storage, Technical University of Denmark, Building 310, 2800 Kongens Lyngby, Denmark; rastu@dtu.dk (R.A.T.); larda@dtu.dk (D.A.); 4Key Laboratory of Eco-Environments in Three Gorges Reservoir Region, College of Environment and Ecology, Chongqing University, Ministry of Education, Chongqing 400044, China; 5College of Chemistry and Chemical Engineering, Henan University, Kaifeng 475004, China; dchanda32@gmail.com; 6State Key Laboratory of Urban Water Resource and Environment, Harbin Institute of Technology, Harbin 150090, China; majunhit@163.com

**Keywords:** electrochemical impedance spectroscopy, tertiary amino groups, anion exchange membrane, zeta potential, pH value

## Abstract

Ion-exchange membranes (IEMs) represent a key component in various electrochemical energy conversion and storage systems. In this study, electrochemical impedance spectroscopy (EIS) was used to investigate the effects of structural changes of anion exchange membranes (AEMs) on the bulk membrane and interface properties as a function of solution pH. The variations in the physico/electrochemical properties, including ion exchange capacity, swelling degree, fixed charge density, zeta potentials as well as membrane and interface resistances of two commercial AEMs and cation exchange membranes (CEMs, as a control) were systematically investigated in different pH environments. Structural changes of the membrane surface were analyzed by Fourier transform infrared and X-ray photoelectron spectroscopy. Most notably, at high pH (pH > 10), the membrane (R_m_) and the diffusion boundary layer resistances (R_dbl_) increased for the two AEMs, whereas the electrical double layer resistance decreased simultaneously. This increase in R_m_ and R_dbl_ was mainly attributed to the deprotonation of the tertiary amino groups (-NR_2_H^+^) as a membrane functionality. Our results show that the local pH at the membrane-solution interface plays a crucial role on membrane electrochemical properties in IEM transport processes, particularly for AEMs.

## 1. Introduction

Ion exchange membranes (IEMs) have been successfully applied in various water and energy systems [1,2]. Desalination by electrodialysis (ED) operating on IEMs for treating industrial effluents and for desalination of sea and brackish water is a long-established technology [3]. More recently, the application scope of IEMs is rapidly expanding to advanced energy conversion and storage systems such as salinity gradient power reverse electrodialysis (RED) [4,5], water electrolyzers [6], CO_2_ electrolyzers, fuel cells, and batteries [7,8,9], among others. In electrochemical water splitting or CO_2_ reduction systems, the limited stability of membranes, particularly the anion exchange membrane (AEM), remains a great challenge [7]. For instance, alkaline water electrolysis operates under high solution pH, which triggers various degradation pathways of the AEMs depending on the type of its functionality [10,11]. In electrochemical CO_2_ reduction, the pH at the interface between the electrode and the electrolyte/membrane increases locally due to OH^-^ generation, which could alter the effective membrane properties in addition to the product distribution during CO_2_ electrolysis [12]. Overall, a controlled pH distribution in a solid polymer electrolyte at the triple-phase boundary in a broad range of electrochemical systems, including fuel cells and electrolyzers, is highly critical as this influences the local reaction environment, reaction pathways/rates, catalytic activities, among others.

Membrane transport properties depend on bulk properties as well as interfacial characteristics, which are both affected by the surrounding electrolyte composition, concentration, temperature, pressure, etc., separately or in combination. Thus, as membrane mass-transport and electrical characteristics play important roles in interpreting the complex transportation mechanism in the IEM system, increasing attention has been drawn recently, which is not only important for industrial requirements but also interesting from an academic viewpoint.

Anion exchange membranes (AEMs) are usually prepared by installing quaternary ammonium groups (-NR_3_^+^_,_ derived from, e.g., N-alkyl pyridiniums or benzyl trimethylammoniums) on various polymer backbones [13], which could fully dissociate at high water contents, while tertiary amino groups, which are also commonly existed in AEMs exhibited deprotonation in the pH range of > 7 [14]. The chemical nature of the backbone, the cationic groups, and the ion exchange capacity all influence the membrane characteristics in solutions with different pH and ionic strength [15,16,17]. Considering the dissociation behavior of these functional groups, it is necessary to study the variation of membrane properties such as membrane and interface resistances under different solution pH. Previous studies on AEMs stability were mainly focused on extreme conditions, i.e., concentrated alkaline solution and high temperature, which caused deterioration of not only the functional groups but also the bulk membrane materials [10,18]. Investigations on the properties of AEMs at lower concentrations of alkaline solution and ambient temperature are more meaningful for practical applications of technologies such as the ones used for the treatment of seawater and brackish water.

Electrochemical impedance spectroscopy (EIS) is a non-invasive technique, which is based on electrical impedance measurements operated with an alternating sinusoidal current with a frequency of 10^−6^~10^9^ Hz [19]. EIS has been successfully applied to study electrochemical processes that occur at different time scales, including multilayer systems characteristics [20], material dielectric and transport properties [21], and electrochemical reaction mechanisms [22]. Since EIS renders the individual assessment of the electrical contribution of each sublayer available, it has been applied to characterize synthetic IEMs [23], composite membranes [24], bipolar membrane structures [25], etc. Previously reported data [26,27] indicate that the resistance across IEMs in aqueous electrolyte solutions is mainly the sum of resistance contributions from three sublayers: the diffusion boundary layer (DBL), the electrical double layer (EDL), and the IEM, with the first two sublayers formed at the interfacial layers of IEM systems. Figure 1 illustrates an AEM in an electrolyte solution operated with an alternating current. The properties of the IEM system are not only affected by the physical properties of the membrane such as fixed charge density and ion exchange capacity, but also by the operating conditions such as flow rate, electrolyte concentration, pH, and temperature. The influence of membrane surface properties and physicochemical characteristics (e.g., morphology and charge distribution) [27] and the effect of operational conditions (e.g., flow rate, electrolyte concentration and temperature) on IEM systems have been widely investigated by EIS [23,26,27,28]. It was observed that the DBL resistance decreased with increasing flow rate, whereas the change in EDL resistance was less significant [27]. When the electrolyte solution concentration was increased from 0.5 to 4.0 M, the resistance of the EDL decreased due to the increase in the screening of the electrical attractions between the fixed charge groups and the counter-ions [26]. While the impact of most operational parameters, e.g., the temperature on IEM resistance, is well described in the literature [29], the influence of electrolyte solution pH on the properties of IEM systems, especially on the interfacial layer properties, remains largely unexplored.

In this study, we systematically investigated the surface chemical properties and electrokinetic characteristics of two commercial AEMs in different electrolyte and pH environments. The effects of structural changes on the membrane properties were studied by using streaming potential and EIS measurements. The variations of membrane resistance, as well as interface resistance and capacitance, were studied in a 0.5 M NaCl solution with a varying pH (2~12). The obtained results are valuable towards the development of highly efficient ion exchange membrane processes for application in water/waster treatment (e.g., ED) as well as energy conversion and storage systems (e.g., CO_2_/H_2_O electrolyzers).

## 2. Materials and Methods

### 2.1. Chemicals and Electrolyte

Sodium chloride (NaCl) and potassium chloride (KCl) (reagent grade, Kermel, Tianjin, China) were used to prepare electrolyte solutions. NaCl solution with a concentration of 0.5 M was used for EIS measurement of the IEM systems. The concentration of KCl in capillaries for Ag/AgCl reference electrodes was 3.5 M. Ultra-pure water (18 MΩcm, Milli-Q, Merck Millipore, Burlington, MA, USA) was used in all the experiments and was degassed before the solution preparation.

### 2.2. Membrane Samples

Two commercial AEMs (i.e., AEM-Type I, AEM-Type II) and 2 CEMs (i.e., CEM-Type I, CEM-Type II) with homogeneous structures were provided by Fujifilm (Fujifilm Manufacturing Europe B.V., Tilburg, The Netherlands). Before the membrane tests, the samples were activated by washing with the test solutions 4 times in 24 h to eliminate the residual chemical solvents and/or impurities.

### 2.3. Characterization of Membrane Samples by FT-IR and XPS

Fourier transform infrared (FT-IR) spectra for the IEMs were carried out by a spectrometer (Perkin Elmer Spectrum One, Perkin Elmer, Waltham, MA, USA) with a total spectral range of 650 to 4000 cm^−1^ and a resolution of 1 cm^−1^. In the XPS test, an X-ray photoelectron spectrometer (XPS-ESCALAB 250Xi) was applied to obtain the surface chemical properties of the membranes, and the procedure was the same as the previous report [27], which had a pass energy of 20 eV with an energy step size of 0.100 eV, and a spot size of 650 μm. XPS data were analyzed by XPS Peak software, and the C1s peak at 284.6 eV was used for calibration of the binding energies.

### 2.4. Ion Exchange Capacity, Membrane Swelling, and Fixed Charge Density

The membrane swelling and ion exchange capacity (IEC) variations in IEMs with electrolyte pH were evaluated according to the following procedure. The surface water on the activated membrane samples (washed with pure water) was removed with a paper tissue, and then the samples were immersed in the solutions of different pH (see Appendix A). The solutions were then stirred for 3 days placed on a shaking plate at a speed of 100 motions/min at room temperature. The samples were thoroughly washed with pure water until neutral pH. Then, the samples were treated by shaking in 0.01 M NaCl for 24 h, and the solution was replaced 3 times during this period to exchange the counter anions with chloride ions (Cl^-^) (for the AEMs) or sodium ions (Na^+^) (for the CEMs) [30]. After that, the samples were washed with pure water until no residual NaCl was retained in the washing solution. 

The IEC of AEM and CEM was obtained by titration with silver nitrate solution (AgNO_3_) and acid-base titration, respectively. The procedures for measurements of IEC and membrane swelling were described earlier [27]. All the measurements were carried out in triplicate along with the blank measurement without membrane. A Metrohm Titrando 905 (Swiss) connected with an ion-selective electrode was used for measurements of IEC in AEM, while a pH combined electrode was used for IEC of CEM. The calculation of IEC, swelling degree, and fixed charge density were shown in the Appendix A, which were referred to in the previous study [27].

### 2.5. Zeta Potential Measurements

Experimentally, the tangential streaming potential (TSP) was measured to interpret the zeta potential of flat membranes. The setup with 2 Ag/AgCl reference electrodes connected to a digital multimeter for measurement of TSP is shown in Figure 2. Two membranes separated by a spacer formed a slit-type channel, which was 2 cm in length and 0.1 cm in width, and the cell height was 0.31 mm for all experiments. The multimeter was used to obtain the electrical potential difference (∆*E_S_*) across the channel. Test solutions of 1 mmol/L KCl, 0.1 M NaOH, and 0.1 M HCl were used to adjust the solution pH. Each test was repeated 3 times. The Helmholtz–Smoluchowski formula was used to interpret the TSP in terms of zeta potential (mV), and the equation was described in our previous study [27].

### 2.6. Electrochemical Impedance Spectroscopy

A 4-electrode setup (shown in Figure 3a) with a Metrohm Autolab potensiostat (PARSTAT302N) was used for EIS measurements with the advantage of eliminating the impedance from the electrode-solution interfaces to help focus on the membrane and interfacial layers [26,31]. The setup was the same as that reported in our previous studies [27,29]. The test membrane was put between 2 reference electrodes with a distance of 1 mm from the tips of reference electrodes to the membrane. The working electrode and the counter electrode were used for applying a small alternating voltage with an amplitude of 10 Mv, which were 2 circular disks of Ag/AgCl with effects of minimizing the water dissociation reactions at the electrodes. EIS measurements were carried out with the frequency of an alternating current varying from 1000 to 0.01 Hz with 50 measurement frequencies per decade. The experimental EIS data were interpreted by fitting the data to an equivalent circuit (Figure 3b), which was reported in our previous study [29]. In order to obtain the pure membrane resistance (R_m_), blank experiments without membrane were carried out to obtain the solution resistance (R_s_), which should be subtracted from the total resistance of solution and membrane (R_m+s_) measured at high frequency. The expression for membrane areal electrical resistance (*R_A_*) is shown in Equation (1):(1)$$R_{A}=R_{m}A$$

Here, *R_m_* is the membrane resistance from EIS measurement and *A* is the surface area of membrane. Measurements were conducted 3 times for each sample. The effective capacitance (*C_eff_*) in association with the constant phase element (CPE) was calculated according to Equation (2), in line with the methods in the literature [27,29].
(2)Ceff=Q1nRCPE1n−1

Here, *R_CPE_* (Ω) is the resistance of the resistor in parallel with CPE, *n* is the CPE exponent and Q (F/(cm^2^S^1−n^)) is the CPE parameter.

## 3. Results and Discussion

### 3.1. Physico-Chemical/Surface Characterization of Anion Exchange Membranes

Membrane characterization for chemical elements on the membrane surface of CEM-Type I, CEM-Type II, AEM-Type I, and AEM-Type II was reported in our previous work [27]. The survey XPS spectra of the four IEMs are shown in Appendix A. It shows that sulfonic acid groups were contained in the two cation exchange membranes (CEM-Type I and II) as fixed charges, while quaternary ammonium groups were in the two anion exchange membranes (AEM-Type I and II) as fixed charges.

The transmittance FT-IR spectra of the AEMs are presented in Figure 4, and the FT-IR spectra of the CEMs are shown in Appendix A in Appendix A. The wideband at 3287 cm^−1^ arose from the stretching vibration of quaternary ammonium salt groups and/or tertiary amino salt groups. The absorption bands at 2917 and 2849 cm^−1^ correspond to the stretching vibration of the aliphatic C-H bonds on the -CH_2_ and -CH_3_ groups [16,32], and the bands at 1330 cm^−1^ correspond to the C-H wagging vibrations [32]. The peaks at 1647 cm^−1^ and 1533 cm^−1^ are due to the stretching vibration of quaternary ammonium groups [33]. The absorption bands observed at 1172 cm^−1^ correspond to N-H in-plane mode [34]. Additionally, the absorption at around 967 cm^−1^ is ascribed to C-N stretching vibration [35]. From the FT-IR spectra, the presence of the quaternary ammonium salt groups were confirmed in the two AEMs, but it was also apparent that the tertiary amino groups may exist in membranes as fixed exchange groups [14].

To check whether tertiary amino groups exist in membranes and to compare the contents of exchange groups between these AEMs, XPS analysis of two AEMs was carried out (the XPS analysis for two CEMs is given in Appendix A). As shown in Figure 5, there were three peaks in high-resolution XPS spectra of N1s of two AEMs. The first peak at 399.48 eV was attributed to the tertiary amino groups on the surface of membranes, while the second peak at 402.36 eV was ascribed to quaternary ammonium groups, and the third peak at 406.24 eV arose from N-O [32,36]. From the XPS spectra of two AEMs, both the tertiary amino groups and the quaternary ammonium groups existed in the membranes as fixed charge groups. The atomic compositions of nitrogen in the two AEM surfaces from XPS analysis are shown in Table 1. It was observed that the contents of nitrogen in tertiary amino groups were higher than those in quaternary ammonium groups on the surface of both AEMs.

### 3.2. Ion Exchange Capacity, Swelling Degree, and Fixed Charge Density of the Membranes 

The changes of IEC and swelling degree of four membranes were measured after immersion in the NaOH or HCl solution for three days. With the increase of pH value, the IEC (Figure 6a) increased and then reached a steady-state for the CEMs, which indicated that the fixed charge groups in CEMs had the lowest dissociation at pH = 2. This was consistent with the apparent p*K*_a_ of 0~1 of sulfonic acid groups [37]. The pKa for quaternary ammonium groups of AEM through theoretical calculation was higher than 15 [38], indicating that the dissociation of quaternary ammonium groups would not be affected by the pH of 2~12 in this study. However, the IEC and fixed charge density (Figure 6b) decreased for the AEMs in alkaline condition, and such a phenomenon was more significant in AEM-Type II than in AEM-Type I, which was probably due to the deprotonation of tertiary amino groups with OH^-^ (pKa of 7~11 [39]) and/or the degradation reaction of quaternary ammonium groups with OH^-^. When the pH was increased from 2 to 3, the IEC and fixed charge density for AEM-Type II increased due to the decrease in the screening effect of the electrical interactions between fixed charged groups in membranes and counter-ions in the solution [26]. C. Iojoiu [40] and Merle [41] pointed out that the quaternary ammonium groups of AEMs in alkaline conditions degraded slowly when the temperature was lower than 60 °C, while it degraded faster when the temperature was higher than 100 °C. From our experiments, it can be hypothesized that the decrease of IEC and fixed charge density was attributed to the deprotonation of the tertiary amino groups and the formation of the charge-neutral freebase form. The difference in the IEC and fixed charge reduction of AEM-Type I and AEM-Type II could be related to the amount of tertiary amino groups (5.4% for AEM-Type I, 6.2% for AEM-Type II). The A.C.% for N element in fixed charge groups of AEMs and S element in CEMs can be calculated from IEC, and the discussion on variations of A.C.% with pH was shown in Appendix A in Appendix A. The membrane swelling degree (Figure 6c) increased for the AEMs, whereas it decreased for CEMs after the immersion of membranes in the NaOH or HCl solution. The observed variation in swelling degree values of AEMs may be related to the alteration of the hydrophilic nature of the membrane matrix, which was consistent with reports from T. Chakrabarty [42] and L. Laín [43].

### 3.3. Effect of Solution pH on Zeta Potential of Anion Exchange Membranes

Zeta (ζ) potential is considered as a key parameter describing the membrane surface charge properties in electrolyte solutions. The measurement of zeta potential on membranes has attracted great attention and is considered the most respected characterization of EDL properties [44]. The zeta potentials of membrane samples were measured after immersion in NaOH or HCl solution for 3 days. The calculated zeta potentials from streaming potential measurements of AEM-Type I and AEM-Type II are shown in Figure 7. The zeta potentials were positive for two AEMs due to the presence of -NR_3_^+^ and -NR_2_H^+^ at the membrane surface. The zeta potential of the two AEMs in the pH range of 7~10 was lower than that in the pH range of 4~7 due to the decrease of the dissociation rate of tertiary amino groups at higher pH. When pH > 11 and pH < 2, there was a dramatic decrease in zeta potential, which could be attributed to the increase in the shielding effect of the electrical attractions [26] when the concentrations of H^+^/OH^-^ in the bulk membrane was higher than the test solution concentration (1 mmol/L KCl) [45]. To evaluate the changes of surface charge distribution after immersion in solutions with different concentrations of H^+^/OH^-^, the membrane samples above were washed with 1 mmol/L KCl solution until no extra H^+^/OH^-^ was detected in the wash solution. After that, the zeta potentials of the membrane samples were measured in 1 mmol/L KCl. The results are shown in Figure 7 (black dashed line and red dashed line). It was demonstrated that the surface charge distribution was not influenced by the treatment with acid, while the zeta potential on the membrane surface was reduced at high pH more significantly in AEM-Type II than in AEM-Type I. This was due to the higher content of -NR_2_H^+^ (Table 1) and higher fixed charge density in AEM-Type II than in AEM-Type I (Figure 6b).

### 3.4. Electrochemical Impedance Spectroscopy (EIS)

#### 3.4.1. Influence of Solution pH on Membrane Resistance 

The effect of solution pH (from 2 to 12) on membrane properties was investigated by EIS on AEM-Type I, AEM-Type II, CEM-Type I, and CEM-Type II (the impedance spectra for AEM-Type II with different pH can be found in Appendix A of Appendix A). The membrane and solution resistance obtained by EIS in 0.5 M NaCl with different pH values at ambient temperature is displayed in Figure 8 (the EIS data and fitting results for AEM-Type I and CEM-Type I are shown as an example in Appendix A of Appendix A). It can be seen that the CEMs had lower resistance at pH > 8 or pH < 4, and the tendency was similar to the solution resistance. This is mainly caused by increased conductivity of the solution in and outside of the membrane at high or low pH. The order of migration rate of the ions^-^ in the solution is H^+^ > OH^-^ > Cl^-^ > Na^+^, which was in line with the finding that the conductivity of CEM and solution varies for the different conditions such as acid condition > alkaline condition > neutral condition.

In addition, the fixed charge groups in CEMs, i.e., sulfonic acid groups (-SO_3_H) could dissociate over the entire pH range, and their dissociation degree was less affected by the solution pH [46], which can be proved by the A.C.% of S element in fixed charge groups in Appendix A of Appendix A. However, when the pH was higher than 8, the membrane resistance of two AEMs increased with the increase of pH, which may be explained in terms of the difference in the deprotonation degree of fixed charge groups in AEMs (the reaction is shown in Appendix A of Appendix A) [46] and the degradation of the quaternary ammonium groups in alkali media (shown in Appendix A of Appendix A). The weakly basic tertiary amino group associated with OH^-^ in solutions became non-ion-exchangeable when solution pH was higher than 7, which may also limit the conductivity of AEMs [41]. In this case, the chemical structures for the two AEMs and two CEMs can be speculated as the diagrams shown in Appendix A of Appendix A. The decrease in membrane resistance of AEMs at pH = 12 and in acidic conditions can stem from two reasons. One was due to the increase of ionic strength, which was caused by the addition of OH^-^/H^+^ ions for adjusting the solution pH and the mobility of OH^-^/H^+^ ions that was about three times higher than that of chloride ions [14]. Another important reason was the increased membrane swelling (Figure 6c) as a result of higher electrolyte sorption of the membrane. The results above were in accordance with those reported by Hosseini [47].

To verify the reversibility of the effect of solution pH on fixed charge groups in AEMs, the AEMs used in Figure 8 at different pH conditions were washed with 0.5 M NaCl until the presence of no extra H^+^/OH^-^ in the wash solution. Next, the membrane resistance was measured in 0.5 M NaCl. The membrane resistances before and after washing are shown in Figure 9. The resistance of the AEMs treated by acid remained close to the original value, while it increased after treatment by an alkali solution. The latter demonstrates that the amount of the fixed charge groups in AEMs decreases in alkaline solution, which could be attributed to the degradation of quaternary ammonium groups at high pH solution. The difference in membrane resistance of the two types of AEMs before and after washing were related to the membrane swelling degree (Figure 6c) and the fixed charge density (Figure 6b) as well as the A.C.% of N element in fixed charge groups (Appendix A).

#### 3.4.2. Influence of Solution pH on Interface Resistances and Capacitance

The effects of solution pH on the electrical resistance and the effective capacitance of interfacial layers in IEM systems are displayed in Figure 10. The resistance of EDL (*R_edl_*, Figure 10a) in CEM-Type I and CEM-Type II increased with the addition of NaOH and HCl while the resistance of DBL declined (*R_d_*, Figure 10b). This was attributed to the phenomena in which the OH^-^ and H^+^ were easily sorbed onto the membrane surface by hydrogen bonding and electrostatic forces, and thus causing more ions in the electrical double layer, thereby increasing the mass transfer resistance. At the same time, the rise of electrolyte concentration in the solution increased the conductivity of DBL, and the migration rates of OH^-^ and H^+^ were higher than Cl^-^, which reduced the charge transfer resistance in the DBL. When the OH^-^ and H^+^ were removed from the membrane and solution, the resistance and capacitance of EDL and DBL for CEMs returned to their original values (Appendix A), which indicated that there was no change in fixed charge groups and membrane structures for CEMs. On the contrary, at high pH (>10), the *R_edl_* (Figure 10a) decreased for the two AEMs. This was probably due to the dissociation changes of the tertiary amino groups and the degradation of quaternary ammonium groups on the surface of AEMs, which reduced the number of charged groups on membrane surface and the number of counter-ions sorbed by electrostatic effect in EDL thus as to reduce the mass transfer resistance. The solution pH did not significantly influence the effective capacitance of EDL (*C_edl_*, Figure 10c) and DBL (*C_d_*, Figure 10d) in the case of CEMs, but it had a non-negligible effect on *C_edl_* and *C_d_* in the case of AEMs. This is in agreement with the changes of surface charge density (Figure 6b) and zeta potential (Figure 7) in AEMs.

The AEM samples were washed with 0.5 mol/L NaCl to neutral, and the changes of interfacial resistance and capacitance vs. solution pH are shown in Figure 11. In accordance with the changes of *R_A_*, the resistance and capacitance in the interfacial layers were essentially unchanged after immersion in acid solutions for three days when compared to the neutral condition, while the *R_edl_* (Figure 11a, dash line) and the C_edl_ (Figure 11c, dashed line) in the electrical double layer decreased significantly. This was consistent with reduction in the amount of charged groups on the membrane surface and the IEC (Figure 6a) as well as the A.C.% of N element in fixed charge groups of membranes (Appendix A) at high pH. As shown in Figure 11b,d, the resistance, and capacitance of the DBL for the AEMs at high pH, which was related to the decrease of charges in the electrical double layer in agreement with our previous report [27].

The resistance variations of interfacial layers with pH were much lower when compared to the changes in membrane resistance (Figure 8, Figure 9, Figure 10 and Figure 11), implying the changes in solution pH plays a significant role in altering membrane resistance. Although the changes in the interfacial layer resistance are low, these small variations can be related to the changes in surface charge distribution over the membranes, requiring further elucidation of this phenomenon.

## 4. Conclusions

The electrochemical properties of two commercial homogenous AEMs and CEMs were studied in electrolyte solutions under different pH conditions. The FT-IR and XPS analysis of the AEMs revealed the existence of quaternary ammonium compounds and tertiary amino groups as membrane fixed charge groups. EIS measurements showed that the resistance of the membrane and DBL in CEMs decreased with the addition of NaOH and HCl, and the EDL resistance increased under alkaline conditions, while all mentioned parameters in AEMs were contrary with those measured in the alkaline condition. It was supposed that there were structural changes in the two AEMs, due to the neutralization of the amino groups (weak basicity) with OH^-^ in solutions. This hypothesis was confirmed by comparing the IEC and fixed charge density among the virgin membranes and the membrane treated with NaOH solutions (0.01 M and 0.001 M) and HCl solutions (0.01 M and 0.001 M). Overall, the experimental results revealed that the local pH had a significant influence on the performances of AEMs than CEMs. The revelation of the pH impact on the electrochemical properties of IEMs paves the way to a better understanding of, e.g., degradation mechanisms in alkaline AEM for water/CO_2_ electrolyzers. Moreover, elucidating the membrane and interface resistance in different solutions is the key in understanding the Ohmic and non-Ohmic losses in systems operating in a different pH environment such as reverse electrodialysis and alkaline fuel cells. This study also demonstrated the potential use of EIS as an advanced, powerful technique to quantitatively evaluate the pH-dependent changes of each sublayer (i.e., DBL, EDL, and membrane in the electrolyte) in IEM systems.

## Figures and Tables

**Figure 1 membranes-11-00771-f001:**
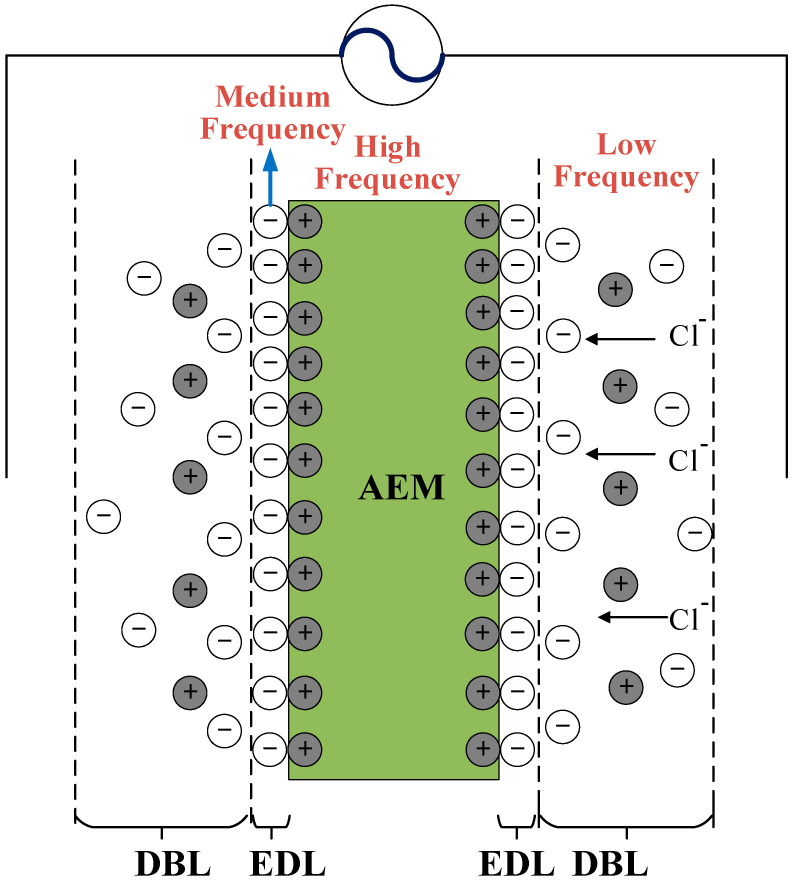
Schematic diagram of an AEM with the diffusion boundary layer (DBL) and electrical double layer (EDL) formed in a sodium chloride solution operated with alternating current. Membrane resistance can be measured at a high frequency. The EDL resistance can be obtained at the medium frequency, while the DBL resistance can be obtained at low frequencies.

**Figure 2 membranes-11-00771-f002:**
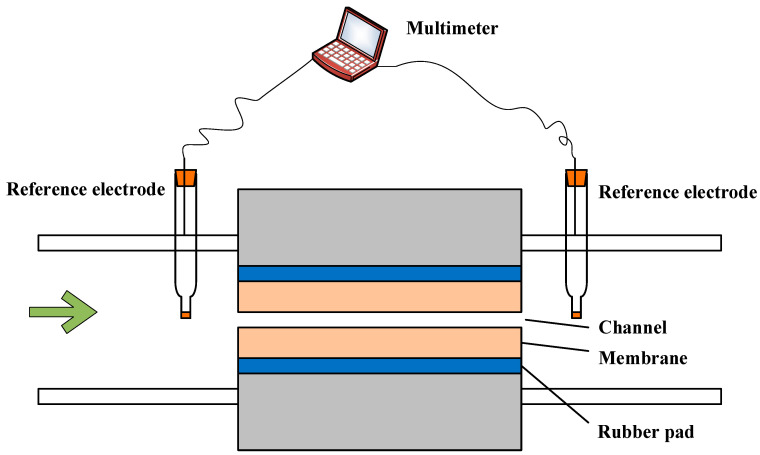
Schematic of the system setup used for tangential streaming potential measurement.

**Figure 3 membranes-11-00771-f003:**
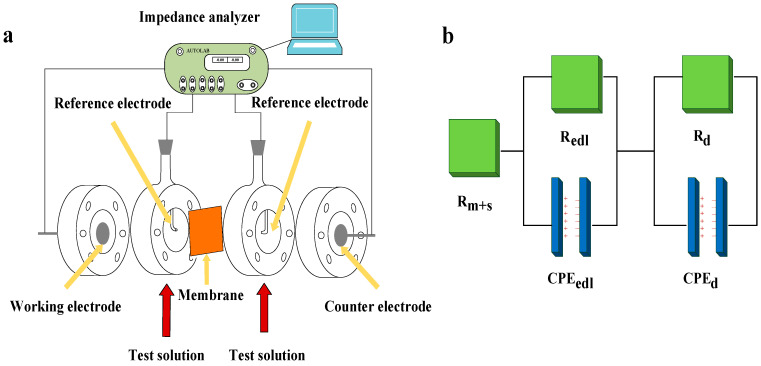
(**a**) Schematic of the setup used for electrochemical impedance spectroscopy measurement and (**b**) equivalent circuit for the data fitting. R_m+s_ is the resistance of membrane and solution, R_edl_ and R_d_ are the resistance of EDL and DBL, respectively, and CPE_edl_ and CPE_d_ are the constant phase element of EDL and DBL, respectively.

**Figure 4 membranes-11-00771-f004:**
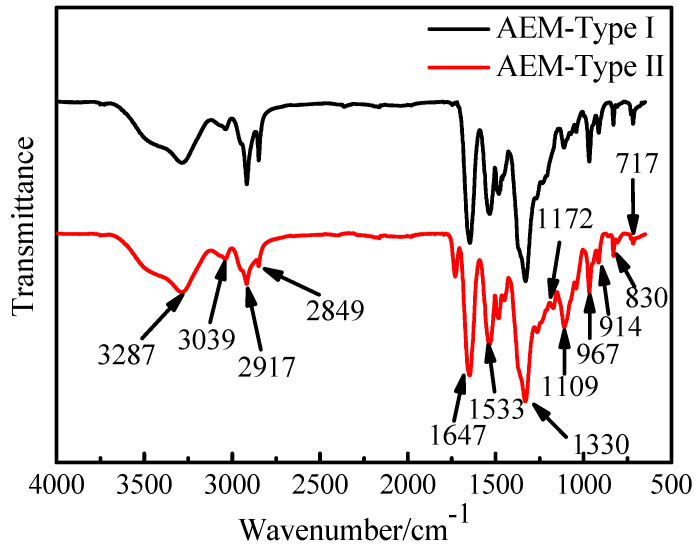
FT-IR spectra of two anion exchange membranes: AEM-Type I and AEM-Type II.

**Figure 5 membranes-11-00771-f005:**
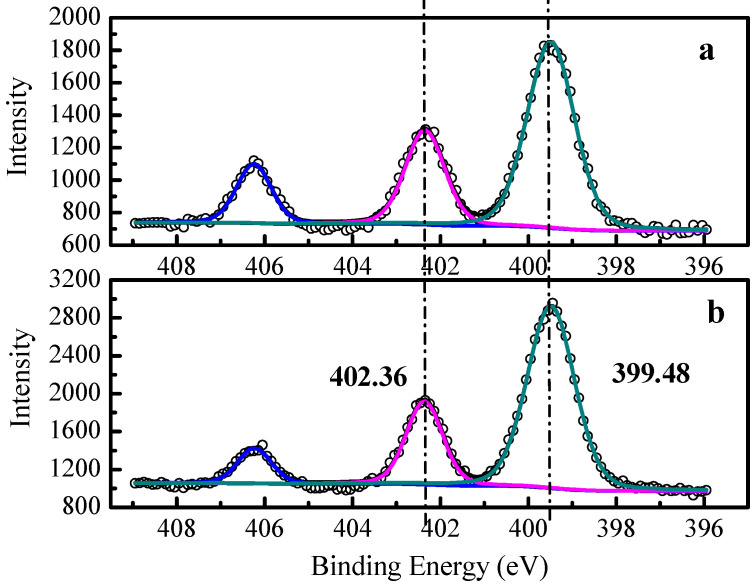
The high-resolution spectra (scattered symbols) and curve fitting (solid lines) of N1s of (**a**) AEM-Type I, (**b**) AEM-Type II.

**Figure 6 membranes-11-00771-f006:**
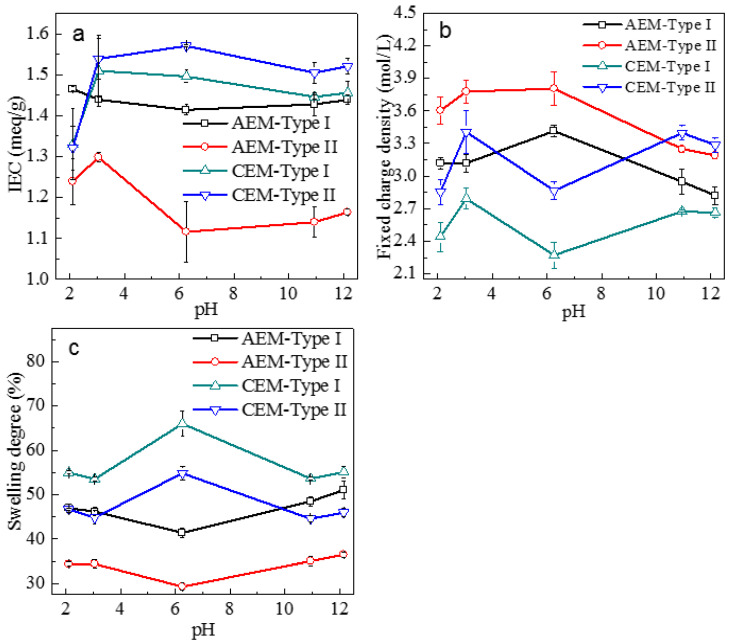
(**a**) Ion exchange capacity, (**b**) fixed charge density, and (**c**) swelling degree of four ion-exchange membranes as a function of pH. NaOH or HCl was used to adjust the pH of the solutions.

**Figure 7 membranes-11-00771-f007:**
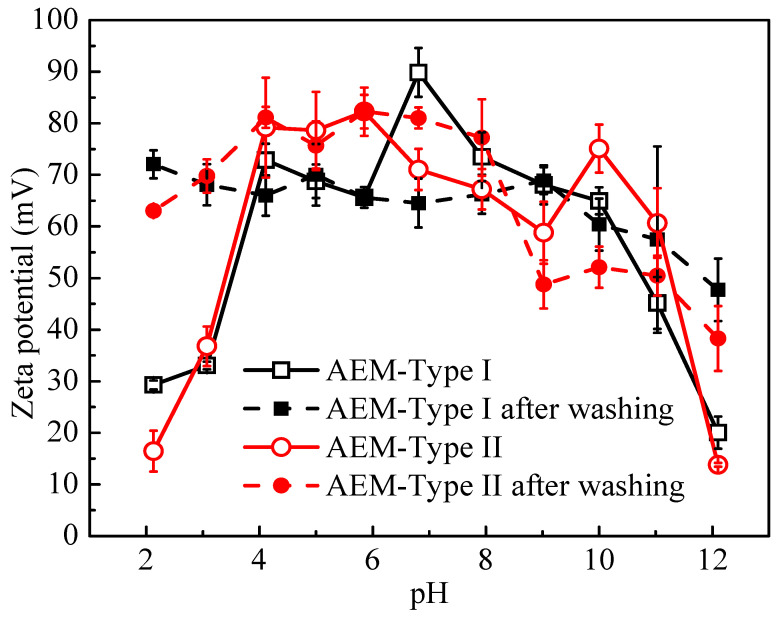
Zeta potential in membrane surface of AEM-Type I and AEM-Type II as a function of pH value in 0.5 M NaCl solution.

**Figure 8 membranes-11-00771-f008:**
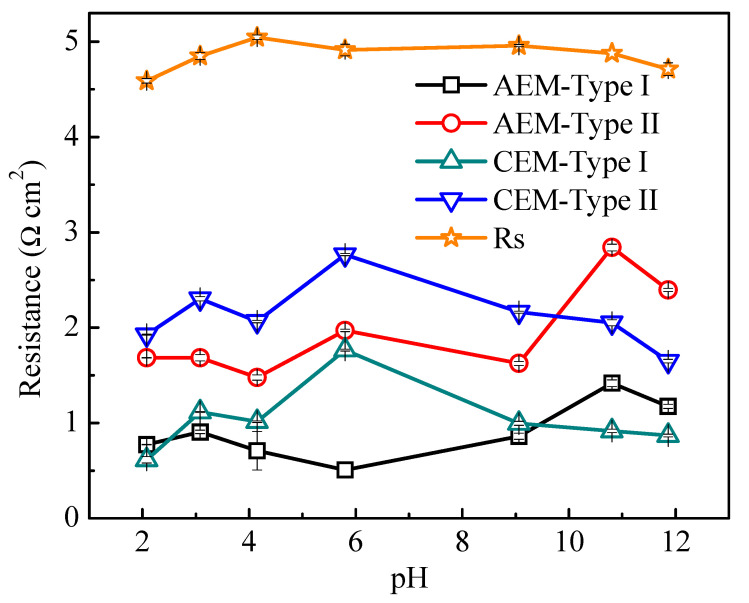
Area membrane resistance (R_A_) of four ion exchange membranes and solution resistance (R_s_) as a function of pH value in 0.5 M NaCl solution.

**Figure 9 membranes-11-00771-f009:**
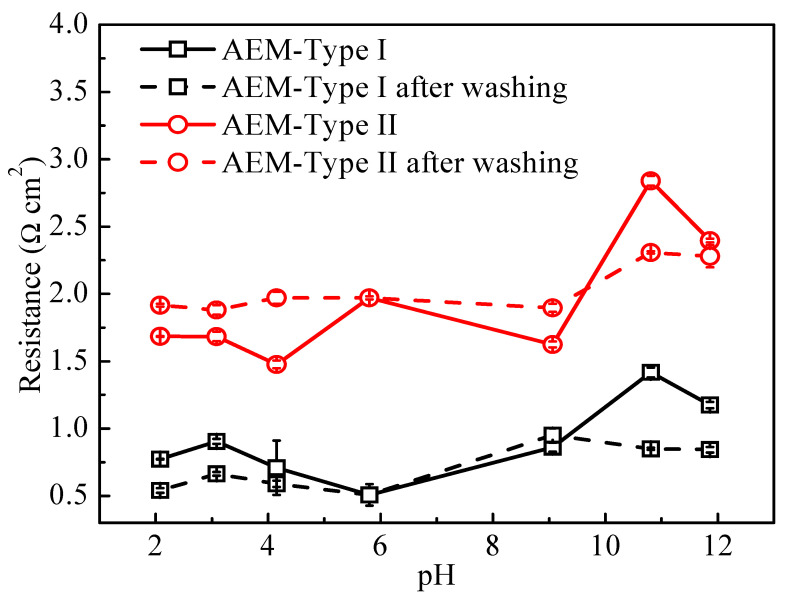
Membrane resistance of anion exchange membranes before and after washing with 0.5 mol/L NaCl solution.

**Figure 10 membranes-11-00771-f010:**
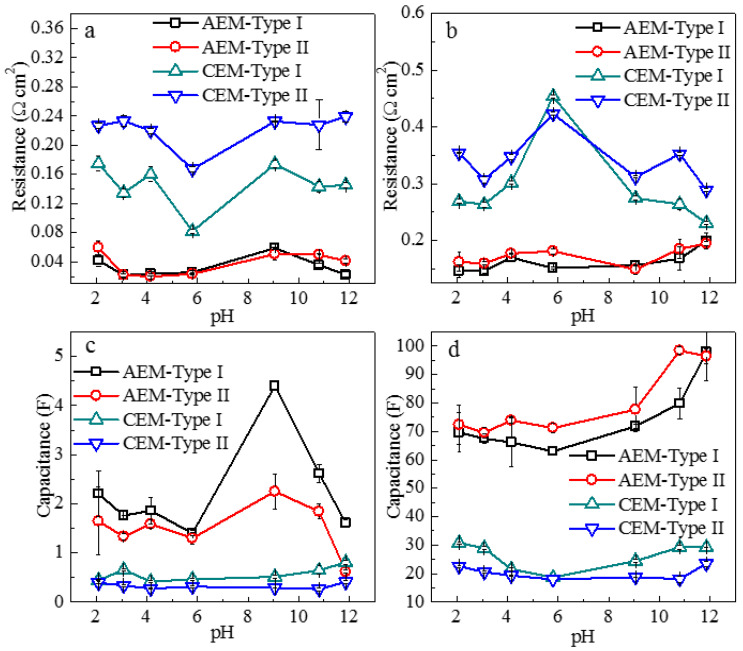
(**a**) Resistance of EDL (R_edl_), (**b**) resistance of DBL (R_d_), (**c**) effective capacitance of EDL (C_edl_), and (**d**) effective capacitance of DBL (C_d_) in four ion-exchange membranes as a function of pH.

**Figure 11 membranes-11-00771-f011:**
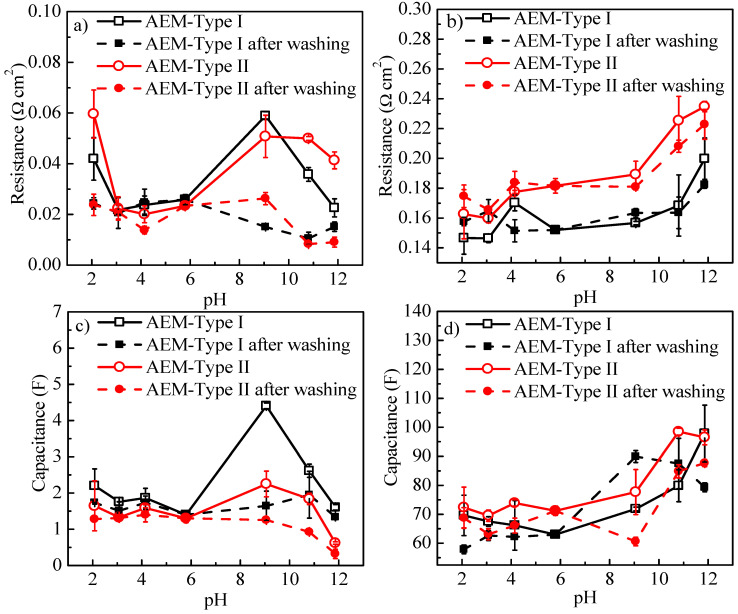
(**a**) Resistance of EDL (*R_edl_*), (**b**) resistance of DBL (*R_d_*), (**c**) effective capacitance of EDL (*C_edl_*), and (**d**) effective capacitance of DBL (*C_d_*) in for anion exchange membranes before and after washing with 0.5 mol/L NaCl solution.

**Table 1 membranes-11-00771-t001:** The atomic compositions of nitrogen in two anion exchange membrane surfaces from XPS analysis.

Membrane	Atomic Composition Percentage, (A.C.%)		
	Total Nitrogen in Membranes (N1s)	Nitrogen in Quarternary Ammonium Groups (-NR_3_^+^)	Nitrogen in Tertiary Amino Groups (-NR_2_H^+^)
AEM-Type I	8.9	2.3	5.4
AEM-Type II	9.3	2.1	6.2

## Data Availability

The data presented in this study are available in this article and Appendix A.

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
