# Peer review of "Studies on Anion Exchange Membrane and Interface Properties by Electrochemical Impedance Spectroscopy: The Role of pH"

_membranes, 2021, doi:10.3390/membranes11100771_

Round 1

Reviewer 1 Report

This manuscript is describing the impact of the pH on some of the properties of two commercial AEMs and two CEMs. The manuscript is well written and the scientific points and statements are clearly present by the authors. The references to the literature are well described and correctly referred. Most of the scientific points are comprehensively presented and clarified.

I have only few concerns about this manuscript:

  1. Although, there are some analysis of the membranes composition e.g. the number of tertiary and quaternary amines, it is not clear to me what is the actual number of functional groups per gram (total IEC). The values about the nitrogen in a.c.% might be supported by other analysis as 1H-NMR or titration methods for instance. Once the total IEC including the IEC fractions of both tertiary and quaternary amines is confirmed by few methods, it can be compared and discussed to all the measurements described in the study.
  2. If the structure of the functional groups is confirmed by the NMR, there will be possible to calculate, refer to scientific studies and experimentally measure (to certain extend) the pKa (pKb) values of those groups. For the CEMs this value is mentioned, but this is not the case for the AEMs. The pKa value for CEM is about 0 to1 (p. 8 line 238). All the measurements are done at pH above 2. This means that the functional groups of the CEM are not in so called H-form (acidic form). Then all the ionic conductivity e. g. resistance measurements are due to a counter ion different then proton. This may explain the lack of any trend in the most resistance vs pH measurements in the case of CEMs. This point however must be clarified for the AEMs.
  3. I do not understand the idea behind the post treatment with 1 mM KCl called “after washing”. On page 9 lines 273 and 278 is mentioned that this treatment will affect the surface charge distribution. However, to certain extend the charges in the bulk of the membranes cannot be excluded. To me a post treatment with KCl will exchange most of the protons (if any) to potassium in CEMs and most of the hydroxyl anions to chloride one in AEMs. This may render any further measurements of the resistance since both ions are much less conductive than protons and hydroxyl ions.
  4. There are some missing data: CEM data are not given and discussed in the figures 7, 9 and 11. “Rs” is not given and discussed in figure 8.

Reviewer 2 Report

The authors investigated the properties of ion exchange membranes depending on the pH of the solution, trying to explain the observed changes from the chemical side, using appropriate methods. The concept of the article is clear and properly implemented.

In order to improve the article, I have the following comments.

  1. I think Fig. 1 should be corrected. Why is the DC source drawn on the top? The figure shows an impedance measurement  of a membrane, that uses an alternating signal. It is to indicate in which frequency ranges the parameters of the membrane and its interface are subject to detection using the technique of impedance spectroscopy. Hence the presence of a DC source is misleading. Also the description "Medium Frequency" is above the DBL layer but it concerns the EDL layer. The purpose of the presence of the little blue arrow is unclear. It seems to me a better solution to place this description above this layer but higher than the other descriptions. 
  2. Line 176, Fig. 3. I have not found any information about the material of the working electrode and the counter electrode. How was the obtained impedance data related to the membrane and the solution only (Fig. 3b) and not to the contribution of the working electrode and the counter electrode? This needs an explanation.
  3. In the supplementary materials, Fig. S2 and S3 show the impedance spectra in the Nyquist format. In such drawings, it is required that the units on the horizontal and vertical axis are the same (then the presence of semicircles and possible distortions can be assessed visually). In this case, they are different, which can lead to wrong judgments. For example, in Fig. S3 we see full semicircles, but it is thanks to the stretching of the vertical axis that in fact the obtained semicircles are quite flattened (CPE character). I recommend changing the units on both axes so that they are of the same length. 
  4. In the supplementary materials, Fig. S2 and S3 show the impedance spectra in the Nyquist format. In such figures, it is required that the units on the horizontal and vertical axis are the same (then the presence of semicircles and possible distortions can be assessed visually). In this case, they are different, which can lead to wrong judgments. For example, in Fig. S3 we see full semicircles, but it is thanks to the stretching of the vertical axis that in fact the obtained semicircles are quite flattened (CPE character). I recommend changing the units on both axes so that they are of the same length.
  5. In Fig. S3b and S3d there is the word 'measurements' (the point description). It should be 'magnitude' or 'modulus'. 

Author Response

Point 1: I think Fig. 1 should be corrected. Why is the DC source drawn on the top? The figure shows an impedance measurement of a membrane, that uses an alternating signal. It is to indicate in which frequency ranges the parameters of the membrane and its interface are subject to detection using the technique of impedance spectroscopy. Hence the presence of a DC source is misleading. Also the description "Medium Frequency" is above the DBL layer but it concerns the EDL layer. The purpose of the presence of the little blue arrow is unclear. It seems to me a better solution to place this description above this layer but higher than the other descriptions.

Response 1: Thank you very much for your comments. We have changed the drawings for the alternating signal applied in the impedance measurement and move the description ‘Medium Frequency’ to the top of EDL layer in Fig. 1.

Point 2: Line 176, Fig. 3. I have not found any information about the material of the working electrode and the counter electrode. How was the obtained impedance data related to the membrane and the solution only (Fig. 3b) and not to the contribution of the working electrode and the counter electrode? This needs an explanation.

Response 2: Thank you very much for your comments. The working electrode and the counter electrode were two circular disks of Ag/AgCl with effects of minimizing the water dissociation reactions at the electrodes. The test membrane was put between two reference electrodes with a distance of 1 mm from the tips of reference electrodes to the membrane. The working and counter electrode were used for applying an alternative current, while the response signals from the two reference electrodes were focused on the membrane and interfacial layers as well as the solution between the tips of the two reference electrodes. To remove the contribution of solution (Rs) between the two tips, blank experiments without membrane were carried out. The pure membrane resistance (Rm) can be calculated by subtracting the solution resistance (Rs) from the total resistance of solution and membrane (Rm+s) measured at the high frequency. In order to make it more clear in the manuscript, we have added the information about the electrodes and test method in line 179~180 and 183~186 as follows. ‘The working electrode and the counter electrode were used for applying a small alternating voltage with an amplitude of 10 Mv, which were two circular disks of Ag/AgCl with effects of minimizing the water dissociation reactions at the electrodes.’ ‘In order to get the pure membrane resistance (Rm), blank experiments without membrane were carried out to obtain the solution resistance (Rs), which should be subtracting from the total resistance of solution and membrane (Rm+s) measured at the high frequency.’

Point 3: In the supplementary materials, Fig. S2 and S3 show the impedance spectra in the Nyquist format. In such drawings, it is required that the units on the horizontal and vertical axis are the same (then the presence of semicircles and possible distortions can be assessed visually). In this case, they are different, which can lead to wrong judgments. For example, in Fig. S3 we see full semicircles, but it is thanks to the stretching of the vertical axis that in fact the obtained semicircles are quite flattened (CPE character). I recommend changing the units on both axes so that they are of the same length.

Response 3: Thank you very much for your comments. I have checked the figures in the supplementary materials and I think the reviewer’s comment was for Fig. S3 and S4. We have accepted the reviewer’s recommendation and changed the units on vertical axis and make the semicircles more significant in Fig. S3.

Point 4: In the supplementary materials, Fig. S2 and S3 show the impedance spectra in the Nyquist format. In such figures, it is required that the units on the horizontal and vertical axis are the same (then the presence of semicircles and possible distortions can be assessed visually). In this case, they are different, which can lead to wrong judgments. For example, in Fig. S3 we see full semicircles, but it is thanks to the stretching of the vertical axis that in fact the obtained semicircles are quite flattened (CPE character). I recommend changing the units on both axes so that they are of the same length.

Response 4: Thank you very much for your comments. This comment is the same as Point 3.

Point 5: In Fig. S3b and S3d there is the word 'measurements' (the point description). It should be 'magnitude' or 'modulus'.

Response 5: Thank you very much for your comments. We have changed the point description to 'magnitude'.

Reviewer 3 Report

The present manuscript describes a well-structured methodology for comparison study using commercial membranes as AEM. Though the detailed information of membranes studied are not available, all performance data and its discussion are consistent, thus I recommend the authors to publish it as it is.

Reviewer 4 Report

In their submission to Membranes entitled "Studies on anion exchange membrane and interface properties by electrochemical impedance spectroscopy: the role of pH", Zhang et al. describe an interesting work of  the effect of pH on two commercial anion exchange membranes. A good characterizataion by means of FT-IR and XPS is also reported. The presented results show that the pH at the membrane-solution interface plays a critical role on membrane electrochemical properties, which has been correctly analyzed. Overall, I recommend the publication of this paper after revising the English style and considering the following points:

1) A scheme describing the chemical structure of the  membranes would be of interest.

add:

The manucript describes an interesting study on the pH effect over the electrochemical properties of anion exchange membranes. The study is well witten and correctly performed. Despite there are other reports where the pH effect has been studied, this study is of wide interest and can be published in Membranes .

Round 2

Reviewer 1 Report

Still missing:

  1. Type and structure of the functional group of the AEM.
  2. pKb (pKa) value of the functional group of the AEM.
  3. Measurements at pHs below the pKa values for CEM and above for AEM.
  4. Explanation of the lack of trends in all the measurements.
  5. What can one learn from the study (in summary).
  6. Beside the answers to reviewers, no much improvement has been done of the manuscript.

Author Response

Thank you so much for all your comments. We have improved the manuscript according to your comments point by point. Please find all the improvements in the attachment.

Round 3

Reviewer 1 Report

Figure S8 is not very informative. Specify R in (-NR3+) is ethyl or methyl? The colors of the postive charges  in the figure S8a are missmatching the colors of the cations given on top. Specify the conter ions.

Author Response

Thank you very much for your careful review and constructive suggestions with regard to our manuscript. Please find the response in the attachment.

This manuscript is a resubmission of an earlier submission. The following is a list of the peer review reports and author responses from that submission.